# Intestinal Homeostasis under Stress Siege

**DOI:** 10.3390/ijms22105095

**Published:** 2021-05-12

**Authors:** Fabiola Guzmán-Mejía, Marycarmen Godínez-Victoria, Alan Vega-Bautista, Judith Pacheco-Yépez, Maria Elisa Drago-Serrano

**Affiliations:** 1Departamento de Sistemas Biológicos, Universidad Autónoma Metropolitana Unidad Xochimilco, Calzada del Hueso No. 1100, CP 04960 Mexico City, Mexico; fabiolagm03@gmail.com (F.G.-M.); avega9713@gmail.com (A.V.-B.); 2Sección de Estudios de Posgrado e Investigación, Escuela Superior de Medicina, Instituto Politécnico Nacional, Plan de San Luis y Díaz Mirón s/n, CP 11340 Mexico City, Mexico; jpachecoy@ipn.mx

**Keywords:** stress hormones, intestinal IgA, intestinal mucus, gut barrier, adrenal glands, enteric nervous system

## Abstract

Intestinal homeostasis encompasses a complex and balanced interplay among a wide array of components that collaborate to maintain gut barrier integrity. The appropriate function of the gut barrier requires the mucus layer, a sticky cushion of mucopolysaccharides that overlays the epithelial cell surface. Mucus plays a critical anti-inflammatory role by preventing direct contact between luminal microbiota and the surface of the epithelial cell monolayer. Moreover, mucus is enriched with pivotal effectors of intestinal immunity, such as immunoglobulin A (IgA). A fragile and delicate equilibrium that supports proper barrier function can be disturbed by stress. The impact of stress upon intestinal homeostasis results from neuroendocrine mediators of the brain-gut axis (BGA), which comprises a nervous branch that includes the enteric nervous system (ENS) and the sympathetic and parasympathetic nervous systems, as well as an endocrine branch of the hypothalamic-pituitary-adrenal axis. This review is the first to discuss the experimental animal models that address the impact of stress on components of intestinal homeostasis, with special emphasis on intestinal mucus and IgA. Basic knowledge from animal models provides the foundations of pharmacologic and immunological interventions to control disturbances associated with conditions that are exacerbated by emotional stress, such as irritable bowel syndrome.

## 1. Introduction

Intestinal homeostasis denotes equilibrium in the face of constant changes resulting from the complex interactions among microbiota and humoral and cellular components of innate and adaptive immunity [1]. These players contribute to maintaining gut barrier function, which is a product of opposite and balanced events that take place at the intestinal epithelium which, in turn, acts as interface between the outside and inner milieu [2]. These “ying-yang” actions include: (i) tolerating the microbiota inhabitants, (ii) allowing transport without causing an immune response to innocuous molecules, and (iii) hampering the entry of enteropathogens and harmful agents with potential inflammatory or toxic activity [2]. The gut barrier comprises four major components. The first is a biological barrier supplied by luminal microbiota, with a critical role in the microbial antagonism that controls the decolonization of enteropathogenic agents. Moreover, a bilateral interaction between microbial and epithelial metabolisms regulates oxygen availability and nitric oxide production. These gaseous compounds drive the balance of microbiota abundance and diversity, resulting in eubiosis, which contributes to homeostasis in the intestinal tract [3,4,5]. The second is a biochemical barrier of mucus that overlays the epithelial cell surface; mucus prevents direct contact between luminal microbiota and the surface of the epithelium to reduce potential inflammatory responses. The third component is a mechanical barrier provided by a polarized epithelial monolayer of several cell types, bonded at the most apical extreme of the paracellular membrane by tight junction proteins. The epithelial monolayer determines gut permeability, namely the rate of flux of molecules across the epithelium [6]. The epithelial monolayer expresses the polymeric immunoglobulin receptor (pIgR) at the basolateral surface; this protein allows for immunoglobulin A (IgA) transport [7]. The final component is an immune barrier comprising gut-associated lymphoid tissue, including immunocompetent cells (T, IgA+ B, dendritic, and IgA+ plasma cells, among others) and humoral effectors such as IgA [6].

Anatomically, functionally, and structurally, the intestinal tract is divided into the duodenum, the jejunum, the ileum, and the colon. The gut barrier components display regionalized distribution in each segment [8]. Microbiota abundance, IgA levels, and pIgR expression increase from the proximal intestine to the colon. Moreover, tight junction proteins involved in the control of permeability of the epithelial monolayer show differential distribution in each gut segment [9,10]. 

The intestinal barrier function can be modulated by stress [11]. Stress outcomes depend on (i) frequency of exposure to a stressor that is single (acute) or repeated (chronic); (ii) duration of exposure to the stressor; (iii) intensity, measured by stress hormones, blood pressure, and heart rate; and (iv) persistence after applying the stressor stimulus [12,13]. Stress inputs result from the bilateral routes of communication between the brain and the intestine through the BGA [14,15]. This structure comprises an endocrine branch via the hypothalamic-pituitary-adrenal axis (HPA), as well as a nervous branch from the autonomic nervous system, including the central nervous system (sympathetic and parasympathetic) and the enteric nervous system [16,17].

Previous studies have explored the impact of stress on microbiota [18], gut epithelial permeability [14,19,20,21], and IgA [22]. This contribution provides the first overall review of the impact of stress on several components of gut homeostasis, with special emphasis on intestinal IgA and the mucus layer. In this review, we discuss the how stress differentially impacts the small intestine and colon due the regionalization of the intestinal immune system [10], and even the parasympathetic branch of autonomous nervous system [23]. Knowledge of the underlying neuroendocrine mechanisms from experimental animal models provides insights for pharmacological and immunological interventions. Novel products that are highly effective at minimal doses, and with minimal or no side effects, may mitigate intestinal disturbances associated with emotional stress, such as irritable bowel syndrome (IBS) and inflammatory bowel disease (IBD) [24,25].

## 2. An Overview of the Biochemical and Immune Gut Barrier Components

Microbiota orchestrates a wide array of events that contribute to homeostasis, including the modulation of oxygen availability in the luminal milieu [26]. In fact, oxygen supply shapes the composition of microbial communities in each region of the intestinal tract, so that higher oxygen depletion correlates with higher abundance of strict anaerobic microbiota in distal rather than proximal segments, as documented experimentally in mice [26]. An anaerobic environment is provided by non-microbial mechanisms that comprise oxidative reactions of epithelial monolayer cells, as seen in germ-free mice [26]. Interplay microbiota-epithelial cell metabolism is supported by the fact that, in colonic cells, oxidative phosphorylation reactions that require high oxygen consumption generate an anaerobic environment that favors the growing of obligated anaerobes; the latter are able to metabolize insoluble fibers to generate fermentation derivatives, with nutritional and modulatory actions beneficial for the host [4]. Disruption of colonic anaerobic metabolism might increase the oxygen supply, favoring the expansion of facultative anaerobic microbes leading dysbiosis [4]. The response luminal milieu under conditions of oxygen depletion is under the control of hypoxia-inducible factors (HIFs) [3]. HIFs are protein transcriptional factors that regulate the expression of a wide array of genes that promote the adaptation to hypoxia, and include erythropoiesis, angiogenesis, and glycolytic enzymes [3]. Metabolic volatile derivatives released by microbiota and intestinal cells, such as nitric oxide (NO), inhibit HIF activity [3]. NO generation results from enzymatic mechanisms via host nitric oxide synthase (NOS) or microbiota metabolism, and by a nonenzymatic reaction between oxygen peroxide and arginine [3]. Inflammatory conditions impair NO synthesis and tetrahydrobiopterin (BH4) availability, a cofactor for NOS [5]. Low NO and BH4 synthesis disrupts antioxidative mechanisms, favors intestinal motility disorders, and impairs NO-mediated vasodilatation [5,27].

Mucus, the prime microbiota habitat, is a prominent biochemical component of the gut barrier; it covers the apical surface of the epithelial monolayer [28]. Mucus is a gel-forming layer secreted by goblet cells that consist of high molecular weight glycoproteins, mucins, which are built up by sugar polymers known as glycosaminoglycans which are attached to a protein backbone via “O” glycosylation. Mucus has a structurally and functionally regionalized pattern so that, in the large intestine, mucus encompasses two layers: a dense inner layer which is firmly attached to the apical face of the epithelial monolayer, as well as an outer layer which is loosely attached to the dense underlying inner layer; the small intestine contains only one loose layer of mucus. The loose outer layer traps microbiota components that penetrate it; hence, it acts as a shield that blocks direct contact of embedded or “stuck” microbes with the underlying inner layer. Moreover, the loose outer layer acts as a gel-forming matrix, where secretory IgA (SIgA) and many others antimicrobial agents are secreted. Unlike the loose outer layer, the dense inner layer is impenetrable to microbiota; hence, it hampers direct contact between luminal microbiota and the epithelial surface, and reduces potential inflammatory responses [28].

The gut epithelium is a single polarized monolayer that acts as a physical barrier and harbors lymphoid populations of so-called intestinal intraepithelial lymphocytes (IELs) [29]. These cells are located at the basal membrane, intercalated, and constrained between the epithelial cells of the gut monolayer. IELs display a phenotypic and functional regionalization so that, in the large intestine, the IELs are both TCD4+ and TCD8+ and express α/β T cell receptor (TCR). In the small intestine, the majority of IELs that express α/β TCR are TCD8+; in addition, IELs that express γ/δ TCR are very abundant [30]. IELs have a key role as keepers of epithelial surveillance to protect against pathogenic agents. IELs also interact with both the epithelial cell monolayer and subepithelial players of gut-associated lymphoid tissue (GALT) to maintain gut homeostasis [29].

The basolateral membrane of the epithelial monolayer interacts with GALT, which comprises immunocompetent cells (T, IgA + B, dendritic, and IgA+ plasma cells, among others) and humoral effectors, such as IgA. GALT comprises two branches: inductive sites such as Peyer’s patches and an effector compartment, represented by the lamina propria. Peyer’s patches are organized lymphoid structures enriched in immunocompetent cells committed to T-dependent IgA generation. This process is determined by cognate interaction between antigen-specific B cells and CD4+ T cells that express a CD40 ligand (CD40L, also termed CD154) that interacts with CD40 on B cells. This ligation triggers activation-induced cytidine deaminase (AID) in B cells that mediates somatic hypermutation (SHM) and class switch recombination (CSR). Along with CD40L, transforming growth factor α (TGF-α) secreted by CD4+T cells is essential for the induction of T-cell dependent IgA class switching [31,32]. In addition, T follicular helper (Tfh), Foxp3+ Tregs and Th17 cells release interleukin (IL)-2, IL-4, IL-5, IL-6, IL-10, IL-13, IL-17A, and IL-21 to enhance the effect of TGF-β on CSR and/or to stimulate clonal expansion of IgA+ B cells [31,33].

The lamina propria, a diffuse lymphoid compartment, is where T-independent IgA CSR occurs by innate pathways. T-independent antigens such as lipopolysaccharide (LPS) activate B cells via Toll-like receptors (TLRs), while polysaccharides trigger B cell activation through B cell receptor (BCR). In addition, B cell stimulating signals are provided by soluble CSR factors, including B-cell activated factor (BAFF) derived from plasmacytoid dendritic cells (pDC), innate lymphoid cells (ILCs), monocytes, and TLR-activated epithelial cells, and a proliferation-inducing ligand (APRIL) released by macrophages and activated T cells [31,34].

After undergoing SHM and CSR at the germinal center of Peyer’s patches, newly developed IgA+ B plasmablasts migrate to mesenteric lymph nodes to reach the gut lamina propria [34]. Transit of IgA+ plasmablasts cells to the lamina propria entails the binding of the ligand α4β7 integrin, which is expressed by IgA+ plasmablasts, with the receptor mucosal addressin cell adhesion molecule (MadCAM) 1, expressed by intestinal epithelial cells. Regionalized homing to the small intestine is guided by an interaction between chemokine receptor CCR9, expressed by IgA+ B cells with chemokine ligand CCL25 derived from epithelial cells; homing to the colon is mediated by the ligation of CCR10 in IgA+ B cells, with CCL28 released by epithelial cells [34,35].

Under the influence of IL-4, IL-5, IL-6, and IL-10, IgA+ plasmablasts mature into plasma cells that synthetize and release dimeric IgA (dIgA) or polymeric IgA (pIgA) [33,36]. Both dIgA and pIgA present at the lamina propria interact with pIgR, a transmembrane protein expressed at the basolateral side of the epithelial cell monolayer. After accomplishing the uptake, pIgR-dIgA and pIgR-pIgA complexes are transported via vesicle-mediated transcytosis. After reaching the apical epithelial surface, pIgR is broken down to generate a secretory component, so that the protein complexes are released as secretory IgA (SIgA) at the lumen [7]. dIgA and pIgR show an expression gradient, lower at the duodenum and higher at the colon [10]. Both pIgR and SC contribute to maintain gut homeostasis [7].

SIgA is an essential effector of intestinal immunity. Indeed, it mediates a staggering array of functions: pathogen clearing and agglutination, toxin neutralization, microbiota metabolite regulation, microbiota growth selection, antigen uptake for antigen-presenting cells, mucosal tolerance, collaboration with IgM and IgG for systemic immune responses, and it acts as an anti-inflammatory antibody [31,37]. The interplay between SIgA and microbiota has a critical outcome on gut barrier function [38]. By excluding some species, most commensal gut microbes induce the low-affinity IgA generation via T-independent mechanisms [39,40]. SIgA exhibits cross-species reactivity based on the interaction of SIgA glycans with similar surface glycans, expressed on even unrelated species [40]. Under inflammatory conditions, gut microbiota is coated with IgA to a greater extent than at a steady state [38]. The propensity of SIgA to coat colitogenic bacteria might contribute to ameliorating the inflammatory response [41]. These IgA-coated microbe complexes interact with the epithelial surface to strengthen the intestinal barrier, increase pIgR production, and downregulate the pro-inflammatory responses [38]. An anti-inflammatory outcome of microbiota-IgA complexes has been documented in in vitro assays conducted in human multicellular three-dimensional cultures composed of epithelial, mononuclear endothelial cells, and fibroblasts [42].

Through these multilevel actions, IgA establishes a crosstalk with microbiota, mucus, the epithelial monolayer, IELs, and subepithelial components of innate and adaptive immune responses to support the barrier function that maintains gut homeostasis [6].

## 3. Overview of Neuroendocrine Stress Pathways

Humoral and cellular components of mucosal immunity are controlled by stress via neuroendocrine pathways [22]. The most prominent endocrine pathway activated by stress results from the hypothalamic release of corticotropic hormone (CRH). This hormone is targeted toward the pituitary to induce the secretion of adrenocorticotropic hormone (ACTH). After entering the bloodstream, ACTH reaches the zona fasciculata of the adrenal gland cortex, where it induces the synthesis of glucocorticoids, corticosterone, and cortisol to be released into blood circulation. Nervous pathways of stress activation are under control of the autonomic nervous system [11], which comprises three branches: sympathetic, parasympathetic, and enteric [17]. The sympathetic pathway of stress encompasses the central activation of the sympathetic branch and ensuing activation of nerve fiber endings that innervate the adrenal gland medulla; the final outcome is the release of stress hormones, including the catecholamines epinephrine (adrenaline) and norepinephrine (noradrenaline) [11]. The vagus nerve is the main parasympathetic innervation of the intestine; acetylcholine serves as the main neurotransmitter. The intestinal vagus nerve comprises anterior and posterior branches. The anterior branch shows prominent innervation at the stomach and the proximal small intestine, whereas the posterior branch joins the ventral celiac branch to innervate the distal duodenum, the jejunum, the ileum, the cecum, and the colon [43]. The vagus nerve delivers inputs that ameliorate inflammation and support intestinal immunity [16].

The ENS is an intricate and expansive network of neurons connected with extrinsic sympathetic and parasympathetic nerves. The ENS is known as the “gut’s little brain” and has an autonomous function but it can modulate and/or be modulated by sympathetic and parasympathetic nerves [17]. Bilateral communication between intrinsic enteric fibers and the vagus nerve has been evidenced by the impact of vagotomy (vagus nerve resection) on vasoactive intestinal peptide 8 (VIP8). Both left (anterior) and right (posterior) vagotomy reduced VIP8 in the duodenum, while right vagotomy increased VPI8 in the colon [44]. VIP8 is a gut-derived neuropeptide involved in the modulation of IgA secretion and IgA+ plasma cell responses [45].

The HPA and the sympatho-adrenomedullary catecholamine system are the paradigmatic and most known pathways of stress. Thereby, corticosteroids and catecholamines have been used as endpoint markers in several experimental stress settings, as discussed in the subsequent sections.

## 4. Stress and Immunoglobulin A

Given its critical role, IgA is a target of experimental animal models to address the outcome of stress on humoral and cellular immune performers of the gut barrier (Table 1). Although there may apparently be discordant findings due to practical issues, they also reflect the multilevel neuroendocrine impact of stress upon complex interactions that IgA maintains with a wide array of gut barrier contributors. 

Downmodulation of IgA due to chronic stress has been documented in the colon of rats under restraint stress prior to middle cerebral artery occlusion (MCAO), in a stroke model [46], in the feces of rats under alternating home/metabolic cage transfer stress [47], in the small intestine of rats exposed to heat stress [48], and in mesenteric lymph node secretions of rats exposed to psychological stress [49]. In addition, reduced IgA levels due to acute stress have been observed in the colon of rats exposed to restraint stress [50]. In murine restraint models, chronic stress can have a downmodulatory effect on IgA levels [51], Peyer’s patches [52], and IELs [53] in the small intestine. In an experimental ulcerative colitis mouse model, induced using 2,3,4-trinitrobenzene sulfonic acid (TNBS), acute stress—water immersion restraint for 4 h— reduced IgA levels, increased inflammation, and caused damage in the colon [54].

Stress-induced IgA-reduction is accompanied by an endogenous increase of glucocorticoids such as corticosterone [46,49,51], and catecholamines such as norepinephrine [51]. Inhibitory effects of chronic stress on both IELs and Peyer’s patch cellularity were unseen in stressed mice treated with RU-486 (a glucocorticoid receptor antagonist), 6-hydroxidopamine hydrobromide (6-OHDA, a synthetic neurotoxic that destroys noradrenergic nerves), or mimicked in unstressed mice treated with dexamethasone (glucocorticoid), and epinephrine (catecholamine) [52,53]. Moreover, in mice with a single exposure to restraint stress for 12 h, acute stress decreased the number of TCD3+CD4+, TCD3+CD8+, and B220+ cells, and increased apoptosis at Peyer’s patches; these changes were reversed by RU-486, whereas 6-OHDA did not exhibit any effect on stress-induced apoptosis [63]. The findings suggest that the inhibitory effects of stress on the IgA response and intestinal cellularity via glucocorticoid elevation may entail apoptosis [63]. Glucocorticoids are anti-inflammatory hormones that can induce apoptosis in lymphocytes [64]. The role of glucocorticoids on stress-induced apoptosis in the intestine is not fully known, but they seem to be synthetized at the intestinal level via T cell activation [65]. As described in mice, the relative mRNA expression of CYP11A1, a key steroidogenic enzyme involved in corticosterone synthesis, showed a gradient of expression—from higher to lower going from the proximal intestine to the distal intestine to the colon—following T cell activation using in situ hybridization [65]. An additional presumable mechanism of stress-induced reduction of intestinal cellularity was ascribed to enhanced cell migration [63]. In the systemic compartment, adrenal stress hormones are potent mediators of cell trafficking [66].

A stringent inverse relationship between IgA downmodulation and stress hormone elicitation is not always seen. For example, acute stress did not affect the corticosterone level but decreased colonic IgA production in rats exposed to acute stress (6 h of immobilization) [50]. Chronic stress induced significant corticosterone elicitation but also an insignificant cecal increase in IgA, as seen in mice under water avoidance stress (WAS) for 1 h a day for 7 days [55].

An elevated IgA response has been related to stress hormone elicitation, given that, in adrenalectomized rats, norepinephrine treatment increased IgA levels, α-chain, and pIgR mRNA expression in the duodenum and the ileum; in adrenalectomized rats, corticosterone treatment increased IgA and α-chain mRNA only in the ileum [60]. These findings suggest that IgA upmodulation is related to the effect of catecholamines on the increase in pIgR-mediated IgA transport [60]. Although their role in IgA upmodulation via enhanced pIgR transcytosis under stress conditions is not fully know, glucocorticoids stimulate pIgR mRNA expression, as found in the duodenum of neonatal rats treated with corticosterone [67]. In experimental stress settings, rats exposed for 2 h a day for 14 days to electric foot shocks showed increased IgA levels in mesenteric lymph node secretions, and IgG in the plasma was ascribed to gut hyperpermeability [49].

Gut permeability reflects the integrity of the epithelial barrier function that denotes the rate of flux of molecules across the epithelium, which can be measured by transepithelial electrical resistance (TEER). Thus, TEER decrease is indicative of increased permeability [68]. TEER was decreased in the jejunum of rats under subacute stress by (i) isolation/limited movement for 24 h, (ii) crowding chronic stress for 14 days, and (iii) a combination of both subacute and chronic stress [69]. Altered expression of tight junction proteins involved in the control of intestinal permeability was found in the duodenum of rats under acute WAS for 4 h [70], the jejunum of rats under subacute stress [69], the colon of rats under chronic WAS [71], and the colon of mice under repeated stress by board immobilization 2 h a day for 4 days [72]. Stress-induced corticosterone elicitation underlies alterations on tight junction protein expression and on gut permeability augment, as found in rats exposed to WAS for 4 h [70], rats under subacute, chronic, and subacute/chronic stress [69], as well as rats under chronic WAS [71]. In unstressed rats subcutaneously injected with corticosterone, there was decreased tight junction protein and increased permeability for low molecular weight macromolecule polyethylene glycol 400 (PEG 400) in the colon, but not in the jejunum, and these changes were reversed by RU-48 [71]. These findings suggest a role for stress-mediated corticosteroid elicitation on increased gut permeability in a region-specific manner.

Although several mechanisms may account the role of stress on down- and upmodulation of the intestinal IgA response, experimental data provide additional insights that reflect the complex interplay among microbiota, IgA, and neuroendocrine players [73]. In experimental settings, stress drives multilevel perturbations between the sIgA response and microbiota (Table 1). In this regard, stress-induced IgA downmodulation goes along with the passage of viable gut bacteria to extraintestinal sites such as mesenteric lymph nodes, the spleen, the liver, and the blood, among others [46,48,50]. Extraintestinal bacterial passage is termed “bacterial translocation”, and refers to the penetration of luminal bacteria via the epithelium into the lamina propria, and then to the mesenteric lymph nodes; stress-induced translocation has been related to an increased IgA response to microbiota, as found in patients with intestinal bowel disease and diarrhea [41]. Stress also triggers the translocation of bacterial derivatives such as LPS, a potent exogenous pyrogen that elicits proinflammatory response via TLR4 [74]. Mice subjected to acute restraint stress (2 h) showed increased colonic expression of innate anti-inflammatory proteins such as heat shock protein 70 (HSP70). In turn, increased HSP70 expression was determined by luminal bacterial colonization enhanced LPS translocation, as well as the elicitation of endogenous glucocorticoid levels [74]. The findings suggest that stress-induced gut barrier weakness allows LPS translocation via endogenous corticosteroid elicitation [46,74].

Stress feeds a vicious circle between IgA and microbiota disturbances that result in intestinal dysbiosis; that is, disruption of growth and stabilization of microbiota that normally colonize the gut lumen. These changes lead to overgrowth of a predominant microbe communities and the loss of diversity [55,56,57,59,61]. In a murine model of subchronic and mild social defeat stress for 10 days, as a model of depression, stress-induced dysbiosis caused significant (up or down) changes in the abundance of fecal microbiota members; these changes correlated with reduced cecal IgA production [56]. Stress-associated dysbiosis might lead to both elicitation of luminal IgA or IgA-coated microbiota that reflect outgoing inflammatory perturbations [55,57,59,61]. In this regard, in mice subjected to chronic WAS for 1 h a day for 8 days, stress in combination with antibiotics exacerbated the dysbiosis, and there were more wall-adhered microbiota and increased cecal SIgA production [55].

As documented in specific free germ (SPF) or germ free (GF) mice, stress-induced dysbiosis stimulated the opening of colonic goblet-cell-associated gaps. Opening goblet cell gaps is inhibited by microbiota-induced Myd88 (a TLR signal protein adapter) activation by blunting the goblet cell response to acetylcholine [75]. Moreover, stress-induced gaps lead to diarrhea, increased bacterial translocation related to an enhanced IgA microbiota response, and more IgA-coated microbiota [59]. Stress-induced dysbiosis in mice has been associated with gut barrier weakness. These findings mimicked the dysbiosis and IgA-coated microbiota increase found in patients diagnosed with IBS who had diarrhea [41,59].

The role of adaptive immunity components on stress-induced intestinal alterations has been addressed in T cell receptor α-chain gene knockout mice (*Tcra^−/−^*) exposed to chronic WAS [61]. This protocol increased the concentration of free IgA in the colon, a phenomenon related to both the loss of bacterial diversity and exacerbation of colitis severity. Stress-induced dysbiosis was strain specific: it occurred in *Tcra^−/−^* C57BL/6 mice but not in *Tcra^−/−^* BALB/c mice. The findings underscore the critical role of T cells in maintaining the diversity and stability of gut microbiota; under stress conditions, defective T cell functions aggravated dysbiosis, reduced microbiota diversity, and increased IgA secretion due to altered gut barrier function [61].

Murine models have addressed the impact of stress on intestinal inflammatory diseases, such as necrotizing enterocolitis (NEC), that cause high morbidity and mortality in premature neonates [57]. Pregnant mice subjected to stress showed reduced fecal IgA and unchanged IgA breast milk levels. Prenatal stress increased IgA-bound microbiota and dysbiosis in female but not male offspring. Female offspring of prenatally stressed dams exhibited more severe colonic tissue damage in a NEC-like injury model compared with offspring with unstressed mothers. These data indicate that murine neonates can inherit a dysbiotic microbiome from dams that experienced stress during pregnancy, and the dysbiosis of neonatal intestinal microbiome contributes to NEC pathogenesis [57]. Table 1 shows the impact of chronic and acute stress models on IgA and mucus as target components in the current of gut homeostasis.

## 5. Stress and the Mucus Layer

The impact of stress on the mucus layer has been addressed in several experimental animal models (Table 1). Mucin-secreting goblet cells are prominent targets of stress modulation through the activation of mast cells as endpoint effectors of CRH release [76]. The outcomes of stress-induced mast cell activation on mucus depletion also entail the response of pro-oxidative and pro-inflammatory components associated with tissue injury [62,77]. This phenomenon has been documented in mast-cell-deficient and wild-type rats subjected to chronic WAS for 1 h a day for 10 days [62]. Unlike mast-cell-deficient rats, in wild-type rats, stress increased ileal and colonic mucin depletion, increased macromolecular permeability, increased bacterial adherence and penetration within enterocytes, favored neutrophil and mononuclear infiltration, and increased myeloperoxidase (MPO) activity and inflammation [62]. A single exposure to immobilization stress for 30 min revealed that, unlike mast-cell-deficient mice, in the colon of wild-type mice, acute stress increased mucin and prostaglandin E2 (PGE2) release [77].

As documented in several murine models, stress has a prominent impact on goblet cell depletion in the small intestine and the colon [72,77,78]. The presumable mechanism of stress-induced goblet cell depletion via the CRH pathway results from gene protein factors that control differentiation and maturation of goblet cells. These cells arise from a common cell progenitor, as documented in the duodenum of neonatal rats that underwent maternal deprivation stress [79]. Moreover, stress-induced goblet cell depletion may result from apoptosis, as found in ileal epithelial cells from rats subjected to WAS for 1 h a day for 5 days; this phenomenon presumably occurs via the CRH receptor pathway [80]. The effect of stress on goblet cell enlargement, as well as on hyperplasia of enteroendocrine cells and mast cells, has been documented in several experimental stress settings in mice and rats [62,72,79]. The underlying mechanism that mediates stress-induced hyperplasia is not fully known, but it might result from stress-induced CRH receptor 1 (CRH-R1) signal, based on duodenal endocrine cells from neonatal rats that underwent maternal deprivation stress [79].

Experimental models of stress, including WAS, cold-restraint stress, and restraint immobilization stress, have addressed the impact of repeated stress on both reactivation and severity of colitis-associated colonic mucin depletion [81,82]. In mice with dinitrobenzenesulfonic acid (DNBS)-induced colitis, stress impact (acoustic-restraint 2 h a day for 5 days) on colitis outcome was addressed. The authors showed that colitis resolution in 6 weeks was reactivated by stress and accompanied by reduced colonic mucin release and increased colon permeability [82]. These findings suggest a role for stress-induced colitis reactivation via activation of previously sensitized CD4+ T cells [82].

Experimental assays have evidenced the prominent role of mast cell activation via CRH release on the stress-induced alterations of goblet cell cellularity and mucin secretion. The HPA and sympathetic pathways were apparently not involved. In this regard, a single exposure to immobilization stress for 30 min elicited similar corticosterone levels in stressed mast-cell-deficient and wild-type mice [77]. In TNBS-induced colitis in rats that underwent repeated short-term cold restraint, stress-induced colitis severity was associated with increased mucin and goblet cell depletion, along with increased MPO expression [81]. Stress-induced colitis aggravation was abolished by prior administration of mast cell stabilizers and cholinergic activation blockers, but neither adrenalectomy nor an adrenergic blocking agent prevented it [81]. These findings indicate that stress-induced TNBS-mediated colitis aggravation by mast cells involves the parasympathetic but not the sympathetic pathway. Despite the prominent role of mast cell activation via central CRH release on stress-induced alterations on goblet cells, several lines of evidence indicate that additional pathways mediated by peripheral CRH release, as well as parasympathetic and sympathetic neurotransmitter release, are involved [62]. This premise is supported, given that stress-induced release of mucin in rats that underwent wrap-immobilization stress for 30 min was abrogated by intraperitoneal pretreatment with atropine (a parasympathetic muscarinic cholinergic antagonist) and bretylium tosylate (a sympathetic adrenergic blocker) and the CRF antagonist α-helical CRF9-41 [83].

There is not always a straightforward association between stress input and reduced goblet cell cellularity. This outcome might reflect complex crosstalk among stress-induced neuroendocrine pathways, microbiota, and components of intestinal immunity. In this context, the upmodulating effect of stress on mature goblet cell density was examined, along with dysbiosis, luminal bacteria adherence, and corticosterone elicitation in antibiotic-treated mice subjected to WAS for 1 h a day for 7 days [55]. Although stress did not alter goblet cell numbers and Muc2 expression, it altered mucin O-glycosylation, decreased mucin adhesiveness, and induced corticosterone elevation and gut hyperpermeability, as documented in the colon of rats subjected to WAS for 1 h a day for 4 days [84]. Data from a model of chronic unpredictable mild stress (CUMS) in rats indicated that CUMS-induced dysbiosis and inflammation were associated with a reduction in the number of goblet cells. These findings provide experimental evidence for clinical trials that claim stress may predispose patients to an increased risk of inflammation-related gut diseases [85].

In neonatal rats subjected to 3 h of maternal separation during 35 postnatal days hyperpermeability, dysbiosis, and a decreased mucin layer in all intestinal regions were observed [58]. In SPF C57BL/6 mice subjected to restraint stress for 2 h a day for 7 days, dysbiosis and the IgA-bound fecal level were increased along with increased opening colonic goblet cell gaps in response to acetylcholine [59,75]. There was increased opening of goblet cell gaps, decreased IgA, and increased *Escherichia coli* translocation to mesenteric lymph nodes in rats exposed to heat stress [48]. Additional experimental settings indicate that increased colonic injury, a reduced number of goblet cells, and colonic IgA content were significantly aggravated in mice treated with TNBS to induce ulcerative colitis and then subjected to water immersion restraint stress for 4 h [54].

Figure 1 summarizes some of the proposed mechanisms underlie the impact of stress on IgA and mucus as targets components in the current of gut homeostasis.

Bovine lactoferrin (bLf) is a multifunctional iron-binding glycoprotein with modulatory actions on neuroendocrine stress components [86]. At present, little is known about whether bLf can affect IgA and the mucin mucopolysaccharide content under stress conditions; thus, we have addressed this issue. bLf was administered for 7 days in the drinking bottle provided to 8-week-old male BALB/c mice that underwent board immobilization stress for 2 h a day for 4 days (unpublished data). As depicted in Figure 2, bLf ameliorated the stress-induced increase in IgA associated with increased permeability and reduced the mucin concentration.

## 6. Conclusions

The findings from experimental trials have provided foundations for natural strategies focused on mitigating stress-associated intestinal dysfunctions related to gut barrier disturbances. These include hormones such as melatonin, which is produced in the brain and derived from the essential amino acid tryptophan [87], as well as probiotics such as *Lactobacillus farciminis* [84]. Additional approaches tested include diets enriched with long chain polyunsaturated fatty acids with anti-oxidant properties, probiotics, and prebiotics (galacto- and fructo-oligosaccharides), which enhance gut microbiota growth [58].

The studies we have discussed in this review highlight that mucus, goblet cells, microbiota, and innate and adaptive immune players such as IgA collaborate harmoniously to maintain gut barrier function. Moreover, regardless of their origins, all these components have a critical role in enduring the stress siege on intestinal homeostasis.

## Figures and Tables

**Figure 1 ijms-22-05095-f001:**
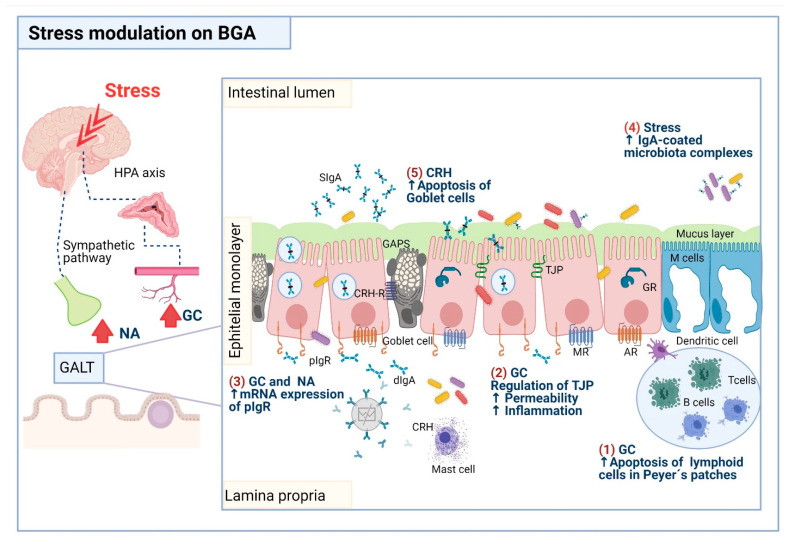
Proposed mechanisms that underlie the impact of stress on components of intestinal homeostasis, including IgA mucin and goblet cells. (1) Lymphoid cells present in Peyer’s patches are susceptible to apoptosis by the actions of glucocorticoids (GC), so the number of plasma cells producing IgA antibodies is decreased. (2) GC disrupt the expression of the tight junction proteins (TJP), resulting in increased intestinal permeability, and concomitantly, bacteria translocation toward the intestinal lamina propria, a phenomenon that can prompt an inflammatory process. (3) GC and noradrenaline (NA) increase pIgR expression promoting the IgA transcytosis and then the release of secretory IgA (SIgA) in the intestinal lumen. (4) Stress-induced IgA-microbiota complexes. (5) Corticotropin-releasing hormone (CRH) secreted by mast cells prompts goblet cell apoptosis and mucus depletion. Abbreviations: AR, adrenergic receptor; MR, muscarinic receptor; GR, glucocorticoid receptor; CRH-R, corticotrophin release hormone receptor. Drawing design by BioRender.

**Figure 2 ijms-22-05095-f002:**
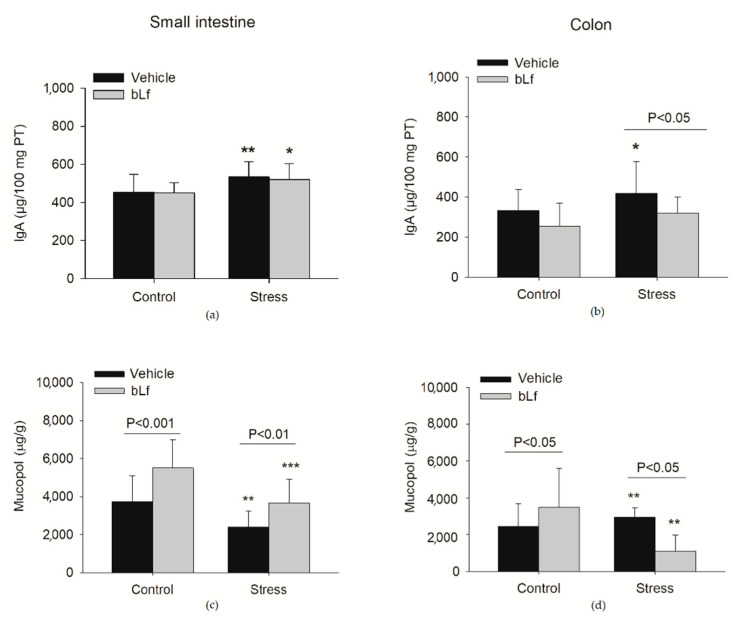
Outcome of stress on IgA and mucin mucopolysaccharides (Mucopol). The concentration of IgA (μg/100 mg of total protein [PT]) in (**a**) the small intestine and (**b**) the colon, and the concentration of mucopolysaccharides in (μg/g sample) (**c**) the small intestine and (**d**) the colon of mice that underwent immobilization stress for 2 h a day for 4 days. * *p* < 0.05, ** *p* < 0.01, and *** *p* < 0.001 indicate column comparisons between stressed and control unstressed groups. The *p* values over the lines indicate comparisons between the paired matched groups.

**Table 1 ijms-22-05095-t001:** Stress and gut barrier components (IgA and the mucus layer).

Chronic Stress Animal Models	
Model	Effect
WAS for 1 h a day for 7 days in antibiotic-treated C57BL/6 mice	↑ IgA cecal content (not significant), ↑ corticosterone, ↑ luminal bacteria adherence, ↑ dysbiosis, ↑ *Clostridium* spp., ↑ mature goblet cell density, ↓ *Verrucobacteria* [55].
Mice exposed to sCSDS 10 days	↓ IgA cecum; IgA and sCSDS levels were correlated, ↓ mRNA IgA response, ↑ cecal dysbiosis [56].
Necrotizing enterocolitis-like murine model in offspring of dams that underwent stress	In offspring from stressed dams: ↓ fecal IgA, ↔ milk IgA. Female offspring of stressed dams: ↑ IgA-bound microbiota, ↑ dysbiosis, ↑ colonic Necrotizing enterocolitis-like injury [57].
Restraint stress for 1 h a day for 7 days in male Fisher rats prior to MCAO	↓ IgA colon, ↑ plasma corticosterone, ↑ bacterial translocation to MLN [46].
Alternating transfer stress in male Sprague Dawley rats (home cage to metabolic cage)	↓ IgA fecal, ↔ fecal and urine corticosterone [47].
Maternal separation stress in neonatal rats	At posnatal day 35 in rats: ↑ intestinal permeability, ↓ intestinal mucin, ↑ dysbiosis [58].
Restraint stress for 1 h a day for 4 days in male BALB/c mice	↓ IgA small intestine, ↑ plasma corticosterone and norepinephrine [51].
Restraint stress for 1 h a day for 4 days in male BALB/c mice	↓ intraepithelial lymphocytes in the proximal small intestine [53].
Heat stress for 2 h a day for 3 days in Sprague Dawley rats	↑ goblet cell gaps in small intestine, ↓ jejunal SIgA, TLR2, TLR4 proteins, ↓ jejunal IL-2, IL-4, IL-10, IFN-γ mRNA, ↑ small intestine injury, ↑ *Escherichia coli* translocation to MLN [48].
Chronic restraint stress for 1 h or 4 h a day for 4 days in male BALB/c mice	↓ IgA+ plasma cells small intestine, ↓ CD8+T and B cells small intestine, ↓ Peyer’s patches cells small intestine [52].
Restraint stress for 2 h a day for 7 days in C57BL/6J SPF mice	↑ fecal IgA-bound to bacteria ↑IgA microbiota response, ↑ opening colonic goblet cells associated gaps, ↓ weight loss, diarrhea, ↑ aerobic bacterial translocation to MLN, ↑ dysbiosis [59].
Restraint for 3 h for 7 days in Wistar rats	↑ IgA levels, ↑ α-chain mRNA proximal and distal small intestine [60].
WAS for 1 h or 1 h a day for 5 days for 12 weeks in T cell receptor α chain gene (*Tcra^−/−^*) knock out mice	↑ IgA microbiota response, ↓ microbiota diversity, ↑colitis, ↑dysbiosis in *Tcra^−/−^* C57BL/6 mice but not in *Tcra^−/−^* BALB/c mice [61].
WAS for 1 h a day for 10 days in mast-cell-deficient ws/ws rats and wild-type control rats	↑corticosterone, ↑ macromolecular permeability, ↑ mucus depletion, ↑ mitochondria enlargement and autophagosomes in epithelial cell layer, ↑ bacterial adherence and penetration into enterocytes, neutrophil, and monocyte infiltration, ↑ mieloperoxidase activity, hyperplasia, and activation of mast cells. No changes in ws/ws rats [62].
Restraint stress for 12 h in male BALB/c mice	↑ Peyer’s patches apoptosis, ↓ TCD3+ cells and ↓ B220+ cells [63].
PS or EFS 2 h a day for 14 days in Sprague Dawley rats	↓ IgA (PS) MLN secretions, ↓ IgG (PS) plasma, ↑ IgA (EFS) MLN secretions, ↑ IgG (EFS) plasma, ↑ corticosterone (EFS) plasma [49].
**Acute stress animal models**	
WIRS 4 h in BALB/c mice that underwent TNBS-ethanol induced ulcerative colitis	In mice that underwent TNBS-induced ulcerative colitis, stress aggravated: ↓ colonic total and SIgA, ↓ IgA serum, ↑colonic mucosa injury, ↓ goblet cells, ↑ IL-6, -8, TNF-α in serum [54].
Restraint stress for 6 h male Wistar rats	↓ colonic IgA, ↔ plasma corticosterone, ↑ bacterial translocation to MLN [50].
Acute restraint stress for 12 h in mice	↓ T and B cells, ↑ Peyer’s patches apoptosis, ↑ endogenous glucocorticoids [63].

Abbreviations: EFS, electric foot shock (physical); EPEC, enteropathogenic Escherichia coli; GFP, germ pathogen free; MCAO, middle cerebral artery occlusion; MLN, mesenteric lymph nodes; PS, psychological stress; sCSDS, subchronic and mild social defeat stress; SPF, specific pathogen free; TNBS, 2,4,6 trinitrobenzene sulfonic acid; WAS, water avoidance stress; WIRS, water immersion restraint stress. ↓ decrease; ↑ increase; ↔ no changes.

## Data Availability

All data generated or analyzed during the present study are included in this published article.

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
