# Peer review of "Intestinal Homeostasis under Stress Siege"

_ijms, 2021, doi:10.3390/ijms22105095_

Round 1
Reviewer 1 Report
The manuscript of Guzman-Mejia, et al. provides an informative review of the effect of intestinal perturbation by stress on inflammatory response. It is a complete review and presents the complicated issues with clarity. I find this a very good review, with only a few small concerns. 1) There is a repeat in the lines between Lines 61-72. Needs to edited. 2) Table 1; Figure 1: While I understand the need for use of abbreviations based on the lack of space, please make sure the authors do not use excessive abbreviations to the point of misunderstanding or confusion. I had to go back and forth many times to figure out all the abbreviations which weren't necessarily consistent with the text or Table 1. For example, the authors use "cort" as an abbreviation of corticosterone in Table 1, but use GC in Figure 1. 3) I am not sure where Figure 2 fits in, and I don't see the mention of it in the text. 4) There are minor grammatical errors that will need to be corrected before publication.Author Response
April 25th, 2021
Dear Prof. Dr. Maurizio Battino
Editor-in-Chief, IJMS
Enclosed please find the thoroughly revised version of the Manuscript ID: ijms-1196698 entitled "Intestinal Homeostasis Under Stress Siege."
All comments and criticisms from the reviewers were found to be extremely valuable to improve the text. Manuscript was throughout revised and all modifications and corrections made by authors as well and those included according to the remarks from the reviewers were highlighted in green.
We hope the current version fulfills the high standards of this prestigious journal and look forward to receiving your last decision.
See please below our Point by Point reply to remarks from the reviewers.
Kind regards,
Maria Elisa Drago-Serrano
corresponding author
mdrago@correo.xoc.uam.mx
Marycarmen Godínez-Victoria
corresponding author
mgodinezv@ipn.mx
Point by point reply (bold)
Reviewer #1 (Remarks to the Author):
The manuscript of Guzman-Mejia, et al. provides an informative review of the effect of intestinal perturbation by stress on inflammatory response. It is a complete review and presents the complicated issues with clarity. I find this a very good review, with only a few small concerns.
1) There is a repeat in the lines between Lines 61-72. Needs to edited.
Thank you so much. In the updated version, repeated lines 61-72 are deleted
2) Table 1; Figure 1: While I understand the need for use of abbreviations based on the lack of space, please make sure the authors do not use excessive abbreviations to the point of misunderstanding or confusion. I had to go back and forth many times to figure out all the abbreviations which weren't necessarily consistent with the text or Table 1. For example, the authors use "cort" as an abbreviation of corticosterone in Table 1, but use GC in Figure 1.
We appreciated so much your careful reading. We punctually reviewed the abbreviations included throughout the manuscript and we deleted all unnecessary acronyms on Figure 1 and Table 1.
3) I am not sure where Figure 2 fits in, and I don't see the mention of it in the text.
Thank you so much. In the current version Figure 2 was referred in the text.
4) There are minor grammatical errors that will need to be corrected before publication.
Thank you. In the current version, all minor grammatical mistakes were corrected
Reviewer 2 Report
Certainly the AA has devoted time and attention to the manuscript. The reason I do not recommend its publication lies in the partial view of a far more complex issue. The Authors in my opinion cannot and should not ignore in their article the role of nitric oxide and oxygen at the gut level.
I would like to bring their attention for example to:
J. Biol. Chem. (2020) 295(30) 10493–10505
Litvak et al., Science 362, 1017 (2018)
www.pnas.org/cgi/doi/10.1073/pnas.1718635115
doi:10.1006/jsre.2000.5862, available online at http://www.idealibrary.com on
Author Response
April 25th, 2021
Dear Prof. Dr. Maurizio Battino
Editor-in-Chief, IJMS
Enclosed please find the thoroughly revised version of the Manuscript ID: ijms-1196698 entitled "Intestinal Homeostasis Under Stress Siege."
All comments and criticisms from the reviewers were found to be extremely valuable to improve the text. Manuscript was throughout revised and all modifications and corrections made by authors as well and those included according to the remarks from the reviewers were highlighted in green.
We hope the current version fulfills the high standards of this prestigious journal and look forward to receiving your last decision.
See please below our Point by Point reply to remarks from the reviewers.
Kind regards,
Maria Elisa Drago-Serrano
corresponding author
mdrago@correo.xoc.uam.mx
Marycarmen Godínez-Victoria
corresponding author
mgodinezv@ipn.mx
Point by point reply (bold)
Reviewer #2 (Remarks to the Author):
Certainly the AA has devoted time and attention to the manuscript. The reason I do not recommend its publication lies in the partial view of a far more complex issue. The Authors in my opinion cannot and should not ignore in their article the role of nitric oxide and oxygen at the gut level.
Thank you for sharing your valuable criticisms. We fully agree that nitric oxide and oxygen have a pivotal role in the intestinal homeostasis. Thereby in the current version, the metabolic interplay of microbiota and intestinal epithelial cells has been devoted as background on base the references suggested from the reviewer and the reference of “Walker MY et al 2018” was included by the authors.
Walker MY, Pratap S, Southerland JH, Farmer-Dixon CM, Lakshmyya K, Gangula PR. Role of oral and gut microbiome in nitric oxide-mediated colon motility. Nitric Oxide. 2018;73:81-88. doi: 10.1016/j.niox.2017.06.003.
Round 2
Reviewer 2 Report
the manuscript is now more informative
Author Response
April 28th, 2021
Point by point reply (bold) Reviewer #2 (Remarks to the Author):
Thank you so much. Due to your comments, the manuscript was substantially improved.